# Workaholism Prevention in Occupational Medicine: A Systematic Review

**DOI:** 10.3390/ijerph18137109

**Published:** 2021-07-02

**Authors:** Thomas Cossin, Isabelle Thaon, Laurence Lalanne

**Affiliations:** 1CHRU de Nancy, Centre de Consultations de Pathologies Professionnelles, Rue du Morvan, 54505 Vandœuvre-lès-Nancy, France; isabelle.thaon@univ-lorraine.fr; 2Department of Psychiatry and Addictology, University Hospital of Strasbourg, 67000 Strasbourg, France; Laurence.LALANNE@chru-strasbourg.fr; 3Department of Psychiatry and Addictology, Medical School of Strasbourg, 67000 Strasbourg, France; 4INSERM 1114, Department of Psychiatry and Addictology, University Hospital of Strasbourg, Fédération de Médecine Translationnelle de Strasbourg (FMTS), 67000 Strasbourg, France

**Keywords:** workaholism, work addiction, prevention, organizational prevention, occupational physician

## Abstract

Introduction: Given the extent of workaholism identified in the literature, it seems essential to consider effective preventive measures. The purpose of this article is to summarize literature data on possible collective and individual preventive measures against workaholism, especially in occupational medicine. Method: We conducted a systematic literature review in accordance with the Preferred Reporting Items for Systematic Reviews and Meta-Analyses guidelines. Results: 155 articles were retrieved in March 2019, but only 15 well-designed studies providing concrete measures to prevent workaholism were included. The various measures were classified using the traditional distinction between three levels of prevention. At the first level of prevention, workaholism can be avoided by implementing a protective organizational culture. The second level of prevention rather focuses on individual training and counselling to address the negative consequences of workaholism. Finally, the third level of prevention combines cognitive and behavioral interventions that enable professional and social reintegration of workaholics. Discussion: This literature review confirms the multifactorial origin of workaholism and the involvement of organizational factors, supporting the necessary contribution of companies in its prevention. This review also reinforces the growing perception of workaholism as a behavioral addiction. Occupational physicians play a key role in this preventive approach as they can influence both working conditions and individual care. The highlighted preventive measures seem to be not only favorable to workaholics, but also to companies. Conclusion: This review provides field tools that can be used at the various levels of workaholism prevention. Nevertheless, intervention studies are required to confirm the effectiveness of the measures presented.

## 1. Introduction

Workaholism was first defined by Oates, who characterized a workaholic as a “person whose need for work has become so excessive that it creates a noticeable disturbance or interference with his bodily health, personal happiness, and interpersonal relations, and with his smooth social functioning” [1]. Since this first definition, this phenomenon has been extensively discussed in the scientific literature, but to date there is still no consensus on its definition and conceptualization [2]. It is therefore not surprising to find disagreements among authors about whether it is a positive [3] or negative [4] behavior, and international classifications (i.e., DSM-5 [5], ICD-11 [6]) still do not recognize workaholism. Workaholics may seem to fit particularly well with work requirements at a time of overall intensification of working conditions [7]. Indeed, one of the most obvious characteristics described in workaholics is a tendency to be highly dedicated to their job and to devote much more time to their work than others do [8], and this over-investment may be socially acceptable and valued within companies [9]. Although the individual part is undeniable in the process of workaholism [10,11], working conditions are also described as a potential etiological factor [12,13]. Therefore, companies may directly contribute to the development and sustainability of the phenomenon.

Nevertheless, workaholism is increasingly considered as a behavioral addiction [14,15,16], which is consistent with the concept initially introduced by Oates [1]. In line with this assumption, an expanding body of evidence is available to support the negative individual and organizational consequences of workaholism [15,16] and its direct and indirect costs to companies [17]. Some authors also argue that the positive consequences of workaholism identified in the literature can be explained by the confusion with the concept of work engagement [18], which has been described as a distinct construct of heavy work investment [19] and commonly considered as a positive form [18]. Considering the high prevalence rates of workaholism documented in the literature (between 5% and 25% according to authors) [20], it is therefore essential to consider possible preventive measures.

Occupational physicians seem to have an essential role in the prevention of this work-specific problem as they can contribute to preventing workaholism both on the individual and professional levels. On the one hand, they perform systematic and regular individual follow-ups of employees during medical consultations, which are particularly conducive for addressing individual work-related issues. On the other hand, occupational physicians can play an essential role in advising companies on occupational health issues.

Preventing workaholism can be difficult considering the centrality of work in modern society [21] and the lack of consistency in the scientific literature on a clear identification of diagnostic criteria. Unlike other types of addictions, complete exclusion from work exposure does not seem reasonable [22]. Furthermore, the lack of social pressure to consider and treat this behavioral addiction and the workaholics’ denial of their problem may complicate the implementation of prevention strategies [23,24]. While many authors have already proposed strategies to manage workaholism [7,25,26,27,28,29], they are mostly based on theorical data, which makes it difficult to have a clear picture of existing and effective preventive measures.

We thus decided to perform a systematic literature review to describe, in the current state of knowledge, the types of individual and collective preventive measures that could be considered to address workaholism by occupational health professionals.

## 2. Method

### 2.1. Protocol

The present systematic review complies with PRISMA guidelines [30,31].

### 2.2. Search Strategy

#### 2.2.1. Information Sources

Seven databases were searched on March 2019 (PUBMED, PsycINFO, Web of Science, BASE, Research Gate, Science direct, Google Scholar via Publish or Perish [32]) using the following combination of terms: [workaholic* OR “work addiction” OR “work craving” OR ergomanie OR boulomanie] AND [prevent* OR treat* OR support* OR control* OR strategy* OR intervention OR occupational* OR addict*]. Bibliographies of the selected articles were then hand-searched for additional references.

#### 2.2.2. Eligibility Criteria

We identified relevant studies based on the following criteria: (1) articles written in English or French; (2) abstracts and full articles available; (3) scientifically peer-reviewed studies; (4) workaholism as the main subject; (5) studies on the prevention of workaholism (i.e., studies focusing on at least one of the three levels of prevention or practical implications of results). If there was any doubt about the abstract inclusion, the study was included for full-text reading.

#### 2.2.3. Study Selection

Our initial search resulted in 3150 studies. After removal of duplicates (*n* = 1210), a first selection was made only based on the screening of titles and abstracts by using the eligibility criteria (i.e., exclusion on the following criteria: other languages, abstracts and/or full papers not available, non-peer reviewed studies, irrelevant subjects, workaholism not as the main subject, lack of prevention considerations) and selected 155 studies for full-text reading. Among the remaining studies, we excluded 51 studies for not matching the review’s subject (i.e., workaholism not as the main subject *n* = 10, studies not focusing on prevention *n* = 41). We then only included studies with a sufficient level of evidence (i.e., controlled trial, longitudinal or semi-longitudinal studies). Therefore, an additional 92 studies were excluded for design reasons (i.e., cross-sectional design, *n* = 66; narrative studies, *n* = 15; reviews, *n* = 6; case reports, *n* = 5). Finally, we added three additional studies after checking the reference lists of the selected articles and included 15 studies in the review as they assessed various impacts of workaholism and provided insights into the prevention of workaholism at the organizational or individual level (Figure 1).

## 3. Results

### 3.1. Data Extraction

We extracted the following items from the included articles: (1) socio-demographic characteristics of the population studied (i.e., countries, ages, gender, education levels, personal workplaces, professional sectors); (2) definition of workaholism; (2) measuring tools; (3) design of the studies; (4) purpose of the studies; (5) main results; and (6) preventive measures.

Table 1 summarizes the study designs, sample characteristics, evaluation criteria, and main findings of the review.

### 3.2. Primary Prevention

The primary level of prevention aims at reducing the incidence of workaholism by minimizing or eliminating identified causes and risk factors. This first level directly involves the work environment or the work situation and therefore the organization and work practices. Eleven studies of the present review described the organization as an essential factor of the workaholism prevention policy [33,35,39,40,41,42,43,45,47,51,53]. To encourage organizations to actively prevent workaholism, eight studies highlighted the need for collective awareness of the phenomenon [35,39,40,41,45,49,51,53].

#### 3.2.1. Work–Life Balance (WLB)

Promotion of a right balance between professional and personal life at the organization level emerges as a determining factor in the primary prevention of workaholism in the studies included in the present review.

The necessity for employees to separate work life from non-work life was showed by Arnold B. Bakker et al. [33] in a study assessing the moderating effect of workaholism on the relation between daily activities during non-working time (i.e., daily time devoted to work-related activities, social activities, and physical exercise after office hours) and daily well-being in the evening (i.e., evening happiness, momentary vigor, and momentary recovery). These results revealed that work-related activities during leisure time were negatively associated with overall well-being, particularly for workaholic employees. Therefore, the authors emphasized the need for organizational policy and implicit standards for limited individual availability and segmentation between work and private life. They also warned against the possible overuse of information and communication technologies (ICT).

Balducci et al. [35] also showed the potential lack of recovery from work induced by workaholism behavior. They performed a study to address the relation between workaholism, job demands, and mental health. Their results revealed that excessive job demands increase the risk of workaholism. They also found a positive relation between workaholism and mental health distress. They argued that a lack of recovery, due to an exclusive investment in work-related activities, may explain mental health distress in workaholic people. Thus, they suggested implementing work-family programs (e.g., programs aimed at developing family-supportive leadership behavior) to alleviate the excessive work investment of workaholics.

Similar results were exhibited by De Bloom et al. [39], who conducted a study to compare the behavioral, cognitive, and emotional effect of vacation (i.e., respectively working hours, ruminations about work, and affective well-being) among compulsive and non-compulsive workers. The authors showed that vacation seemed to be a period of relief for both types of workers, but especially for workaholics on the emotional level. The authors thus suggested improving organizational policies on vacation (i.e., “the right to and sufficient opportunities for regular and longer time off work”) to lead the way to a healthier lifestyle.

More recently, T. Huyghebaert et al. [40] also stressed the need for a healthy WLB as part of the prevention of workaholism. They demonstrated that working excessively significantly mediates the effects of workload on both work-family conflicts and on the lack of psychological detachment from work. Their results mainly imply that workaholism can be prevented by setting clear organizational segmentations and norms (i.e., “work schedules including breaks and specific hours at which employees should leave the office, making sure they take time off work—long and frequently enough”).

Interestingly, instead of focusing on individual consequences (i.e., the employee’s well-being), Falco et al. [41] used a theorical model to assess the potential organizational consequences of workaholism. They assessed the impact of workaholism on job performance and sickness absenteeism via physical and psychological restraints. These results highlighted the importance of impaired physical or psychological health on poor job performance and sickness absenteeism among workaholic employees. Because of these negative consequences, the authors suggested implementing an organizational culture primarily aimed at balancing the work and private spheres to prevent workaholism.

One other suggestion to maintain a WLB was made by Avanzi et al. [42], who assessed the mediating effect of workaholism on the relation between organizational identification (i.e., the perception of oneness or belongingness to an organization) and well-being among employees. They found that organizational identification is usually positive for workaholic employees and thus increases their well-being. They also showed a curvilinear relation between organizational identification and workaholism, which means that when individuals identify themselves too strongly with their organization, they may develop dependency on their work. Therefore, to maintain a healthy WLB they suggested fostering organizational identification by providing rooms for extra-work activities.

#### 3.2.2. Job Demands (JDs) and Job Resources (JRs)

##### Job Demands

Andreassen et al. [43] particularly highlighted the role of working conditions (i.e., demand-support-control model [54]) on workaholism. They found that JDs and JRs were respectively positively and negatively associated with workaholism. They suggested that workaholism could be prevented by changing workload expectations within companies. Balducci et al. [35] refined this perspective as they demonstrated that, while excessive JDs increase the risk of workaholism, workaholic behavior did not influence JDs over time. Thus, they recommended monitoring JDs and ensuring that they remain manageable during the usual working time. De Bloom et al. [39] also revealed that, while vacation may be an excellent opportunity to relieve the workaholic symptoms, compulsive workers tend to return to their old habits more quickly than other employees after returning to work. Thus, they suggested that reducing JDs and overtime, particularly after resuming work, could extend the positive effect of vacation (i.e., reduction of workaholic behavior) on workaholism. These results are in line with T. Huyghebaert et al. [40], who confirmed that making reasonable organizational JDs seems to be a key factor for preventing workaholism and its potential negative consequences.

##### Job Resources

Hakanen et al. [45] pointed out the workaholic’s lack of JRs in a study assessing the association between workaholism and other types of employee well-being (i.e., work engagement, job satisfaction, burnout) with various types of job crafting (i.e., increasing structural and social resources, increasing challenging demands, and decreasing hindering demands). They showed that workaholism was positively associated with an increase in structural resources, challenging demands, and burnout. As they also found that workaholics did not seek social resources, the authors advised to foster managerial support and feedback and to encourage a less hardworking attitude. Balducci et al. [35] suggested providing social resources at work as they perceived them as potential moderators able to reduce the influence of JDs on workaholism. Deficiency in JRs was also underlined by Andreassen et al. [43] and T. Huyghebaert et al. [40], who respectively proposed to increase social support for workaholics (i.e., increased task variety, more staff, more time to plan work, and increased teamwork) and to provide sufficient JRs (i.e., tools to prioritize and delegate, communication, and performance feedback).

Yulita et al. [47] moderated precedent results as they showed that JRs, used in secondary prevention, can foster workaholic behavior in specific organizational contexts. They performed a study to assess the relation between workaholism and psychological safety climate (PSC, i.e., senior management commitment to stress prevention, priority for psychological health versus productivity imperatives, organizational participation and involvement in managing psychological health risks, and organizational communication on psychological health issues). They showed that high PSC had a beneficial effect as it decreased workaholism. They also found that, in companies with poor PSC, some specific JRs (i.e., instrumental JRs: emotional and cognitive resources that help to get the job done and achieve work goals) could be used by workaholics to foster their detrimental behavior, as they may be able to work harder and to achieve their goal of avoiding guilt and anxiety. Their results imply that PSC could operate to temper the possible detrimental effects of instrumental resources on workaholism and should be developed within organizations. Furthermore, interventions aimed at providing instrumental JRs should be used with caution with workaholic employees, depending on the level of perceived PSC. To develop PSC, they suggested developing management training in value-based leadership and leadership for psychological health and positive motivation.

#### 3.2.3. Organizational Reward Systems

As they respectively pointed out the negative individual and organizational consequences of workaholism, T. Huyghebaert et al. [40] and A. Falco et al. [41] suggested not rewarding such kind of behavior. A. Falco and colleagues [41] particularly suggested using reward systems (i.e., career, salary raise) based on working smart rather than working hard.

#### 3.2.4. Exemplary Role of Managers

T. Huyghebaert et al. [40] highlighted the essential role of managers in the prevention of workaholism, as they promoted the exemplary role of managers in encouraging co-workers to have a balanced life. De Bloom et al. [39] also advised supervisors to adopt an adequate behavior on vacation and time at work, to act as role models for their co-workers. Conversely, Balducci et al. [35] cautioned against the detrimental role that managers could play by promoting a “workaholism culture” through their own behavior.

#### 3.2.5. Involvement of Occupational Physicians

The role of occupational physicians is mainly based on prevention and on advising employers on occupational health issues. Their role in the prevention of workaholism may therefore seem essential. Nevertheless, the study by Falco et al. [41] is the only one to point out this role in the primary prevention of workaholism by suggesting that occupational physicians could suggest interventions aimed at changing the work environment.

In sum, workaholism can be prevented by promoting an organizational culture with a balance between professional and private life via an active policy staging clear organizational segmentations and norms [33,35,39,40,41,42], by changing organizational expectations related to workload [35,39,40,43], by providing sufficient job resources [35,40,43,45,47], by using the managers as role models [35,39,40], and by implementing a reward system that does not encourage workaholic behavior [40,41].

### 3.3. Secondary Prevention

The secondary level of prevention intends to reduce the prevalence of workaholism through early detection and individual care. This secondary level is firstly based on the identification of workaholic behavior. Seven of the studies included in the present review explicitly recommended recognizing workaholism as a bad behavior and suggested systematic screening in the workplace [39,40,41,45,49,51,53]. Three studies specifically advised distinguishing work engagement and workaholism as different forms of heavy-work investment [45,51,53], as they showed that each form could respectively lead to good and bad individual and organizational consequences. Two studies provided details on how to perform such screening. They respectively suggested observing the employee’s work habits [40] (e.g., excessive hours spent at the office, late-night work-related emails) and using a dedicated cross-culturally valid scale [40] (e.g., Dutch Work Addiction Scale, DUWAS) or identifying potential workaholics based on individual and professional characteristics [41] (e.g., self-efficacy, neuroticism and hours worked, position held).

#### 3.3.1. Emotional Intelligence (EI)

The main advice related to individual prevention found in the present review is the development of individual EI (i.e., self-awareness, self-regulation, social-awareness, relation management).

In a diary study based on the Effort-Recovery Theory, C. Van Wijhe et al. [48] assessed the role of negative emotions in the process of recovering from work among workaholic and non-workaholic employees. They found a stronger negative effect of negative emotions at the end of the workday on workaholic employees who tended to spend more time at work and had fewer recovery experiences (i.e., restoration of resources depleted as a result of a working day) in the evening. They also found that daily recovery experiences in the evening had a positive impact on the next morning’s emotions, beyond the effect of sleep quality. From a practical standpoint, as they found that negative emotions relate to perseverative cognition (i.e., continuation of work concerns) after work and hamper recovery experiences for workaholics, they argued that time management training may help regulate excessive behavior (i.e., to gain control over their time schedule by setting realistic goals and prioritizing tasks). The same authors [49] also investigated the potential long-term causes and consequences of this lack of recovery in another study assessing the mediating effect of workaholism on the relation between performance-based self-esteem (i.e., gain or maintaining of self-esteem through good role performance) or enough continuation rules (i.e., continuing to work when they felt they had not done enough) and burnout. They found that performance-based self-esteem is related to the cognitive component of workaholism (i.e., working compulsively), and that enough continuation rules foster workaholism at the behavioral and cognitive levels (i.e., working excessively and compulsively). They also found a reciprocal positive relation between working compulsively and exhaustion. Considering the precursors of workaholism (i.e., basing one’s sense of self-worth and work persistence on one’s performance) and its interrelation with burnout, they confirmed that using time management training can be useful for workaholic employees (i.e., helping employees to set realistic goals and to delegate responsibilities so that they can better cope with work-related stress). Similarly, Andreassen et al. [43] and Huyghebaert et al. [40] respectively suggested providing stress management training and developing self-management skills.

The negative affect experienced by workaholics was outlined by Clark et al. [50]. They found an association between workaholism, negative emotion, and work–home conflicts. In their study, anxiety, disappointment, and self-assurance at work emerged as the prominent mediators in the work model (i.e., work-to-home conflict), whereas anger, disappointment, and joviality at home emerged as the prominent mediators in the home model (i.e., home-to-work conflict). Based on the work-to-home results and to help employees cope with anxiety and disappointment at work, the authors suggested developing self-regulation intervention (i.e., developing feedback seeking, proactive behavior, emotional control, and social competences). To address the negative affective states due to workaholism, A. Falco and colleagues [41] proposed training programs aimed at developing individual psychological resources at work (e.g., self-esteem, resilience, active coping style).

Finally, the potential effect of self-regulation strategies has been specifically considered by M. E. L. Zeijen et al. [53] in a study assessing the mediating effect of two self-management strategies (i.e., self-observation, self-goal setting, self-reward, and self-punishment strategies) on the relation between workaholism and expansive job crafting (i.e., seeking challenges, seeking structural and social resources). Their result highlighted that workaholism was related to all expansive job crafting behaviors through self-goal setting and was also associated with self-punishment without mediating effect. As they found no mediating effect of self-observation on the relation between workaholism and job crafting, they suggested motivating workaholic employees to use self-awareness techniques. Balducci et al. [35] also pointed out the potential interest of self-awareness for workaholic employees.

#### 3.3.2. Relaxation Techniques

Relaxation techniques were also presented as a useful way to regulate negative emotions [48], to reduce perseverative cognition during after-work hours [41], and to extend the benefits of vacation on the workaholic’s behavior and recovery from work [39]. The authors suggested that relaxation can mainly be achieved by engaging in structured relaxation techniques, such as practicing muscle relaxation [39] or mindfulness meditation [39,41].

#### 3.3.3. Counseling Based on Self-Validation

Counseling based on self-validation was recommended by Andreassen et al. [43] to help workaholics validate other self-related aspect than work (i.e., spiritual, transcultural-existential, social-cultural, familial, and physical self). This result was in line with other studies recommending the promotion of physical and social activities to prevent ruminations about work-related issues and more generally to enhance recovery and well-being [33,41,42,49]. This recommendation of self-validation as a useful intervention conforms to the primary level of prevention aimed at maintaining a healthy balance between professional and personal life. Some authors reminded of the need for human resource professionals to stimulate workaholics to disengage [33,41,42,49] and to provide a structured work environment [40] for workaholic employees.

#### 3.3.4. Career Counseling

One other individual aspect found in this review is the use of career counseling to help workaholic employees to find a work they enjoy or work they perceive as highly meaningful [43]. A. Mäkikangas et al. [51] showed this potential benefit by investigating the relation between workaholism and job mobility over time. They found that workaholism was associated with a relative stability over time, but that decreasing levels of workaholism were largely explained by voluntary job changes. From a practical standpoint, they suggested that encouraging voluntary job mobility with coaching should help employees find a job that fits them best in terms of well-being and should decrease workaholism behavior.

#### 3.3.5. Meditation Awareness Training (MAT)

As mentioned above, structured relaxation techniques can be based on mindfulness meditation [39,41]. Some authors interestingly reported that the use of MAT may also improve individual EI through self-awareness [53] and self-regulation [40].

W. Van Gordon et al. [52] highlighted the interest of meditation awareness in workaholism prevention in a non-randomized controlled trial aimed at evaluating the efficacity of MAT (i.e., engaging a full, direct, and active awareness of experienced phenomena that is spiritual in aspect and maintained from one moment to the next) as a specific intervention for workaholism. Their results demonstrated significant improvement in control-group participants in terms of levels of workaholism, job satisfaction, psychological distress, work duration, and work engagement without any decline in job performance. They concluded that MAT can be a practical and cost-effective workaholism intervention as it reduces self-attachment and goal-oriented work. MAT may also be used for improving work-related well-being and work effectiveness as a whole.

#### 3.3.6. Involvement of Occupational Physicians

The role of occupational physicians in the secondary prevention of workaholism is poorly substantiated in the present review. Only the study by Mäkikangas et al. [51] supports that occupational health professionals can contribute to the screening of workaholic employees by discriminating between “good” and “bad” forms of heavy work investment.

In sum, secondary prevention of workaholism can be implemented by developing EI [35,40,43,48,49,50,53], relaxation techniques [39,41,48], self-validation [33,35,39,41,43], and career counseling [43,51]. MAT seems to be a particularly effective way to prevent workaholism [39,40,41,52,53] as it mainly aims at regulating withdrawal symptoms, dysphoric moods, myopic focus on reward, salience, life priorities, and impatience.

### 3.4. Tertiary Prevention

The tertiary level of prevention extends prevention to the field of rehabilitation as it seeks to promote professional and social reintegration after illness. This stage of prevention is aimed at helping workaholics recover from negative consequences and rebuild healthy work behaviors to avoid relapse. However, it seems difficult to distinguish between measures that fall within the scope of tertiary prevention and therapeutic care measures as these measures also aim to treat chronic disabilities.

#### 3.4.1. Rational Emotive Behavior Therapy (REBT)

Cognitive behavioral therapies (CBTs) are presented as the main treatment for workaholic employees in the present study.

A. Falco et al. [41] and Balducci et al. [35] both suggested using cognitive behavioral interventions to help workaholics substitute their irrational beliefs about work with more realistic beliefs. Consequently, workaholics can be helped into reducing their work-related worries, setting boundaries between work and home, and taking time for non-work activities. More specifically, C. Van Wijhe et al. [48] argued that REBT may be an adapted treatment for workaholism as it can be used to uncover the irrational beliefs that underlie the workaholic’s negative affective state and to teach them how to counteract maladaptive emotions and irrational cognitions. The same authors [49] confirmed their suggestion in another study as they showed that REBT could change rigid cognition and could act on the performance-based self-esteem and enough continuation rules identified as part of the workaholic’s irrational beliefs (i.e., demandingness, which refers to absolute ideas of how oneself or others should behave and/or low frustration tolerance, which represents intolerance to discomfort, difficulties, and frustration).

#### 3.4.2. Relaxation Training

Relaxation training seems to be another cognitive behavioral method in the tertiary prevention of workaholism. C. Van Wijhe et al. [48] suggested that relaxation could increase awareness of tensions and help workaholics undo their negative emotions by stimulating positive feelings. More particularly, muscle relaxation techniques are presented by A. Falco et al. [41] as useful to reduce the tension attributable to the negative affective states related to workaholism.

#### 3.4.3. Workaholics Anonymous (WA)

Balducci et al. [35] mentioned self-help groups such as WA as an opportunity to recover from workaholism through multi-step programs.

#### 3.4.4. Other Preventive Measures

To prevent occupational exclusion and to help workaholic employees remain in employment, all preventive measures of the first and second levels presented above should also be considered at this stage of prevention.

#### 3.4.5. Involvement of Occupational Physicians

Unlike the two other levels of prevention, the role of occupational physicians at the tertiary level of prevention was better highlighted in the present review. Effectively, C. Van Wijhe et al. [48] argued that REBT may be useful for the treatment of workaholics by health professionals and A. Falco et al. [41] specifically suggested that occupational physicians could implement, either personally or jointly with psychologists or psychotherapists, cognitive behavioral interventions.

In sum, the tertiary level of prevention for workaholic employees can be implemented by referring workaholics to support groups such as WA [35] and using cognitive behavioral supports [35,41]. Using REBT [48,49] or relaxation training [48] seems to be particularly relevant. Primary and secondary prevention measures should also be considered to help workaholic employees remain in employment.

## 4. Discussion

### 4.1. Global Discussion

The present review includes studies performed between 2012 and 2018. Although the concept of workaholism was developed more than 40 years ago, the recent nature of these studies may reflect past difficulties in reaching a consensus on this phenomenon. It is interesting to note that the studies included in our review studied people with an overall high level of education. It is therefore difficult to generalize our results. Although several studies have suggested that occupations with high levels of autonomy may be more vulnerable to workaholism [55,56], a recent metanalysis concluded that there is no significant relation between workaholism and education level [57]. Most studies in our review were conducted in European countries. As literature data reports a high prevalence of workaholism in “Eastern countries” [58], we expected a higher number of studies with a focus on prevention conducted in these countries. Nevertheless, to date, the impact of potential cultural differences needs to be further assessed.

Despite the apparent lack of consensus in the scientific community, we observed consistency in the definition of workaholism among the studies of our review. Overall, all authors supported the addictive nature of workaholism. They perceived it as the association of excessive and compulsive work behavior, with loss of control, despite the negative impact on other areas of life. This definition fits rather well with the criteria for behavioral addiction defined by U. Goodman [59].

Our review supports the multifactorial nature of workaholism, as it confirms the common hypothesis that organizations can play a crucial role in its primary prevention. As workaholism may seem beneficial to companies, employers may be reluctant to invest in its prevention. The role of occupational physicians then seems essential as they must warn employers against the harmful human and organizational consequences of workaholism. Awareness of the stakes involved in this phenomenon can thus encourage companies to adopt preventive measures.

### 4.2. Practical Implications

Overall, the literature review suggested a combination of individual and organizational factors in the emergence of workaholic behavior. Thus, it is important to act on all three levels of prevention. Acting only on secondary or tertiary prevention would restore the individual’s state of well-being and strengthen the resources acting as a protective factor against workaholism, but excluding primary prevention measures could generate a vicious circle where the recovered individual is reintegrated into a deleterious organizational context.

Current working methods—particularly through the permanent access to work provided by ICTs—blur the boundaries between the various spheres of life [60]. The benefits of an organizational culture that promotes a work–life balance have been highlighted in the literature [61]. It seems important to underline that WLB is not defined as equality between work and non-work domains but rather as the extent to which individual effectiveness and satisfaction in each domain are consistent with their own values at a given moment in time [62]. Therefore, organizational cultures that promote WLB could be particularly suited to the prevention of workaholism, as workaholics typically experience work–life imbalance [63,64,65]. Based on our results, the concern of Bakker et al. [33] on the use of ICTs seems to be justified as there is growing literature evidence that the use of technology for professional purposes after regular working hours has negative implications for WLB [66]. This statement seems all the more true not only in a context of organizational telepressure [67,68], but also in the prevention of workaholism, as researchers have shown that telepressure can be self-inflicted by workaholic people [69]. Organizations should therefore take into consideration these potential risks induced by ICTs, especially for workaholic people. This concern also seems to be a societal matter as the French public authorities introduced into the French Employment Code the “right to disconnection and the implementation by the company of measures to regulate the use of digital tools to ensure compliance with rest and leave periods as well as respect of personal and family life” [70].

Companies also need to be particularly alert of their managers’ behaviors, among whom the prevalence of workaholics has been shown to be high [56]. According to Bandura’s social learning theory (i.e., individuals are influenced by the behavior of significant others [71]), managers can indirectly encourage other employees to develop workaholic tendencies [2]. They must therefore be aware that their own behaviors can have both a negative and positive influence on their co-workers. One interesting lead for companies is to train managers on “ethical” leadership [72], as this kind of leadership seems to be an effective way to prevent workaholism through role modelling [73].

In line with the study by Falco et al. [41] included in our review, the implementation of specific rewards related to esteem and support should be considered by companies, as they may have a protective effect against workaholism [74]. This hypothesis is based on the assumption that narcissistic tendencies could make workaholics crave rewards and recognition within the workplace and push them to always put in extra effort [75]. It seems to be relevant as many authors found that workaholic people also present a strong need for achievement [33,76], and the correlation between narcissism and achievement has been well demonstrated [77,78].

With regard to the JDR model, organizations must make reasonable JDs as their negative relation with workaholism is widely supported in our review and can be linked with the workaholic’s performance-based self-esteem and perfectionist personality [74,79]. The lack of influence of workaholism on JDs shown by Balducci et al. [35] may seem surprising in our review. Indeed, such an idea contrasts with the dominant view, as workaholic employees are known to seek more work [80] it is reasonable to expect them to create pretexts for working, hence unnecessary self-imposed JDs [81]. On this point, further research appears necessary. The role of JRs seems to be more contrasted based on our results, and companies should be cautious in allocating them. On the one hand, JRs may be beneficial in primary prevention as they moderate the effect of JDs on the development of workaholism. On the other hand, JRs can act in a contradictory way in secondary prevention because workaholics may use them to reinforce their inappropriate behavior. In this case, as assumed by Yulita et al. [47], specific organizational culture seems to impede this relation [82].

Companies can also provide individual support to workaholic employees as part of secondary prevention. Workaholics are known to report low job and career satisfaction, low job performance, and high turnover intention [83]. Individual career counselling can thus help them find meaning in work and learn to appreciate it, even if it means changing jobs [84]. A growing interest for employee assistance programs (EAPs) has recently been observed. Such programs are designed to help individuals resolve acute but modifiable behavioral health problems [85]. These programs were initially created to help alcoholics deal with their behavior in the work environment; they may therefore be particularly well fitted for addiction disorders. Moreover, it has been shown that workaholics are referred to an EAP for concerns other than this specific issue [76]. This implies that human resources professionals and career counsellors must be trained and aware of the individual and professional challenges of workaholism.

Although organizations play a crucial role in preventing workaholism, it is also essential to act on the individual aspects involved in this behavioral addiction. Workaholics are known to experience work-related negative emotions and have difficulty in managing them [86,87]. This statement is in line with findings from our review and strongly suggests a lack of EI among workaholics, which is defined as “the capacities of knowing and managing one’s emotions, motivating oneself, recognizing emotions in others, and handling relationships” [88]. Workaholics should therefore be helped in becoming aware of their emotional disorders, and specific self-regulatory individual trainings should be suggested.

Encouraging workaholics to engage in extra-work activities seems logical as they focus exclusively on professional activities [84,89]. Given the Effort-Recovery model, activities using the same internal resources as those required at work should be avoided to be able to recover [90]. However, engaging in extra-professional activities is not enough. Detaching oneself psychologically and mentally disengaging from work-related thoughts and concerns is important [91]. The recovery process depends on both behavioral factors and, above all, on psychological factors that must be addressed first [92,93,94].

Mindfulness-based interventions (MBIs) and CBT therefore seem to be particularly relevant. They both show promising results in the field of addiction, whether it is on substance use disorders [95,96] or behavioral addictions [97,98,99,100]. So far, Van Gordon et al. [52] are the only ones to have evaluated the direct benefit of MAT on workaholism. Nevertheless, other studies showed that MBIs may have beneficial effects on workaholism as people learn to self-regulate their emotions [101], to achieve better recovery through psychological detachment [102], and to adopt a better WLB [103]. However, as evidenced in our review, REBT can be useful to act on the workaholic’s irrational beliefs that drive them to excessively and compulsively invest in their work. Some authors suggest focusing on irrational beliefs of performance demands and failure that workaholics self-inflict on themselves [104,105]. CBT has also been shown to have beneficial effects as it enhances emotional awareness and regulation [106,107].

Before being able to give advice to workaholics, it is necessary to identify them. Given the current debate on workaholism, there is no official recommendation for its systematic screening. Occupational physicians should therefore be vigilant about specific warning signs and initiate secondary screening if necessary. French occupational physicians therefore have at their disposal validated tools (DUWAS [108]. WART [109]) to establish diagnosis. Unfortunately the BWAS scale [110], presented in our review as an essential tool, has yet to be approved in France. In addition to their advisory role, occupational physicians may rely on other medical specialists. The contribution of addiction physicians or psychiatrists seems particularly indicated for workaholism, whether in terms of diagnosis or therapeutic care, especially as literature data reports the existence of co-addictions [20] and associated psychiatric disorders among workaholics [111]. Finally, in France, occupational physicians may also call on occupational disease centers, which are composed of specialists in various fields of occupational health and can provide expert advice on individual work situations. This multidisciplinary management allows occupational physicians to implement measures adapted to the workaholic employee’s health status, and thus promotes recovery.

There is no doubt that individuals must face certain immutable changes in working conditions. However, the ability to adapt may vary from one person to another and some may gradually develop a negative relation with their work. It is at the heart of prevention to provide strategies to adapt working conditions to individuals. This is not necessarily incompatible with current work requirements. On the contrary, in the present review, the prevention of workaholism seems to be beneficial for both companies and employees.

### 4.3. Limitations

Our study was not without limitations. Firstly, only studies in French or English languages were selected. Secondly, our method involved an initial reading based only on the title and abstract, using our selection criteria to identify potential studies. This method proved to significantly reduce the number of studies included for a full reading, and it is likely that some of these studies had potential useful preventive measures for workaholism and were screened out at this stage. Indeed, the lack of focus on prevention in the titles or abstracts does not allow us to know with certainty that there were no preventive measures described in full articles. Furthermore, except for one study, the objective of the selected studies was not to directly assess the validity of specific measures to prevent workaholism. Therefore, the prevention measures proposed in our review were mainly deduced and their actual effectiveness has yet to be assessed. In addition, the only study evaluating a specific workaholism prevention method was a non-randomized controlled trial with a small sample size (i.e., 50 participants). Finally, despite the empirical results revealing no significant differences between workaholism and education level, it seems important to note that “blue-collar” workers are under-represented in our review and that it would be interesting to study this specific aspect in greater depth.

## 5. Conclusions

Based on current scientific knowledge, the present review highlights essential elements that can be used at various levels of workaholism prevention. Even though this study focused on prevention at the occupational medicine level, we believe that these results can be useful to all field stakeholders who may be confronted with workaholism. Although the concept of workaholism has been scientifically debated for 45 years now, it is necessary to provide field tools considering the potential magnitude of the phenomenon and the associated harmful consequences. It is therefore essential to validate effective preventive methods, even though they will need to be reassessed as knowledge evolves. We hope that the present review will contribute to promote research with rigorous methodologies to assess preventive measures against workaholism. This would later allow for new systematic reviews that could confirm or invalidate some of our results with greater scientific validity. This research was conducted in March 2019, so it would be particularly interesting to supplement it by incorporating the literature on workaholism from the past two years.

## Figures and Tables

**Figure 1 ijerph-18-07109-f001:**
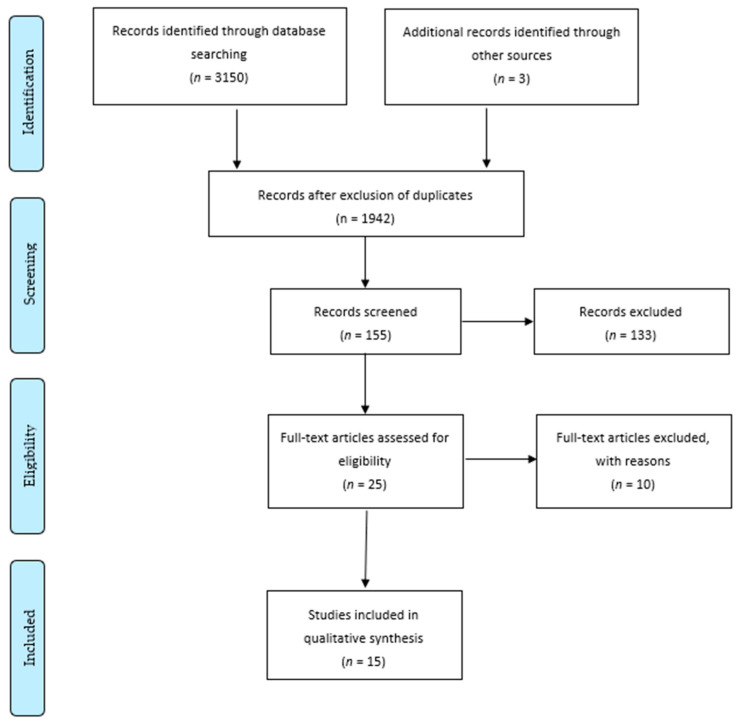
PRISMA flow diagram.

**Table 1 ijerph-18-07109-t001:** Study designs, sample characteristics, evaluation criteria, and main findings of the review.

Authors, Years	Sample	Design	Definitions	Diagnostic Tools	Investigations	Preventive Measures	
Bakker A.B. et al. (2012) [33]	85 participants from Germany Undefined jobs (67% with university degrees) Undefined sectors	Diary study Nine consecutive days One measure a day	Schaufeli et al., 2009 [34]	WART compulsive subscale	Association between, workaholism, extraprofessional activity, and recovery from work	Primary prevention	WLB
						Secondary prevention	Self-validation
						Tertiary prevention	
Balducci C. et al. (2016) [35]	508 participants from Italy Nurses; physicians; administrative staff and technical staff National health care service	Longitudinal study Two-wave design One-year follow-up.	Schaufeli et al., 2009 [34] Ng et al., 2007 [36] Scott et al., 1997 [37] Spence and Robbins, 1992 [38]	DUWAS	Association between workaholism and JDR	Primary prevention	WLB; JDs; JRs; role model
						Secondary prevention	Self-validation; EI;
						Tertiary prevention	CBT; WA
De Bloom et al. (2014) [39]	54 participants from the Netherlands Heterogenous jobs (technicians; associate professionals; legislators; physicians; clerical support; service and sales workers) 53% with college or university degrees Undefined sectors	Longitudinal study Eleven-wave design Nine-week follow-up	Schaufeli et al., 2009 [34] Ng et al., 2007 [36] Scott et al., 1997 [37]	DUWAS	Association between workaholism, vacation, and recovery from work	Primary prevention	WLB; role model
						Secondary prevention	Self-validation; relaxation; MAT
						Tertiary prevention	Relaxation
Huyghebaert T. et al. (2018) [40]	287 participants from France Managers (general; medical; paramedical; social; administrative; technical) Health care sector	Longitudinal study Two-wave design Three-month follow-up	Schaufeli et al., 2009 [34] Ng et al., 2007 [36]	DUWAS	Association between workaholism, JDR, and psychological detachment from work	Primary prevention	WLB; JDs; JRs; role model; reward system
						Secondary prevention	EI; MAT
						Tertiary prevention	
Falco A. et al. (2013) [41]	322 participants from Italy Undefined jobs Undefined educational levels (45.9% white-collar, 52.8% blue-collar) Mechanical engineering company	Longitudinal study Two-wave design 15-month follow-up	Schaufeli et al., 2009 [34]	DUWAS	Association between workaholism, job performance, and sickness absenteeism	Primary prevention	WLB; reward system
						Secondary prevention	Self-validation; MAT
						Tertiary prevention	CBT
Avanzi L. et al. (2012) [42]	140 participants from Italy Teachers Educational sector	Longitudinal study Two-wave design Seven-month follow-up	Schaufeli et al., 2009 [34]	WART compulsive subscale	Association between workaholism and organizational identification	Primary prevention	WLB; organizational identification
						Secondary prevention	
						Tertiary prevention	
Andreassen C.S. et al. (2017) [43]	1308 participants from Norway Nurses Health care sector	Longitudinal study Two-wave design Two- or three-year follow-up	Andreassen et al., 2014 [44]	BWAS	Association between workaholism and JDR	Primary prevention	JDs; JRs; self-validation
						Secondary prevention	EI; career counseling
						Tertiary prevention	
Hakanen J.J. et al. (2017) [45]	1877 participants from Finland Dentists Health care sector	Longitudinal study Two-wave design Four-year follow-up	Schaufeli et al., 2009 [34] McMillan et al., 2006 [46]	DUWAS	Association between workaholism and job crafting	Primary prevention	JDs; JRs
						Secondary prevention	
						Tertiary prevention	
Yulita I. et al. (2016) [47]	392 participants from Malaysia Police staff—35% with bachelor’s degrees or above Police department	Longitudinal study Two-wave design Four-month follow-up	Schaufeli et al., 2009 [34]	DUWAS	Association between workaholism, PSC, and JDR	Primary prevention	JDs; JRs; PSC
						Secondary prevention	
						Tertiary prevention	
Van Wijhe C. et al. (2012) [48]	118 participants from the Netherlands Heterogenous jobs (scientific; administrative; management; consultants; engineers) 60.2% with bachelor’s degrees Heterogeneous sectors (Undefined)	Diary study Five consecutive days Three measures a day	Schaufeli et al., 2009 [34] Ng et al., 2007 [36] Porter, 1996 [23] Mudrack et al., 2004 [11]	DUWAS	Association between workaholism, negative emotions, and recovery from work	Primary prevention	
						Secondary prevention	
						Tertiary prevention	Relaxation; REBT
Van Wijhe C. et al. (2014) [49]	191 participants from the Netherlands Scientific staff; support staff—92.6% with college or university degrees Educational sector (University)	Longitudinal study Two-wave design Six-month follow-up	Schaufeli et al., 2009 [34] Scott et al., 1997 [37]	DUWAS	Association between workaholism and rigid personal beliefs	Primary prevention	
						Secondary prevention	
						Tertiary prevention	REBT
Clark M.A. et al. (2014) [50]	340 participants from the U.S.A. Undefined jobs Undefined education level; undefined sectors.	Longitudinal study Two-wave design Undefined follow-up	Schaufeli et al., 2009 [34] Scott et al., 1997 [37] Porter, 1996 [23]	WART compulsive subscale	Association between workaholism, negative and positive emotions, and work–home outcomes	Primary prevention	
						Secondary prevention	EI
						Tertiary prevention	
Mäkikangas A. et al. (2013) [51]	463 participants from Finland Managers; 90% with academic degrees Heterogenous sectors (manufacturing; information; real estate; rentals; service; trade; financing; insurance; public administration; education; health care; public relation; public transport)	Longitudinal study Two-wave design Two-year follow-up	Schaufeli et al., 2009 [34] Ng et al., 2007 [36] McMillan et al., 2006 [46]	DUWAS	Association between workaholism and job change	Primary prevention	
						Secondary prevention	Career counseling
						Tertiary prevention	
Van Gordon W. et al. (2016) [52]	50 participants from England Undefined job (46.5% with university degrees) Undefined sectors	Nonrandomized controlled trial Eight-week intervention Pre-post intervention analysis Three-month follow-up	Andreassen et al., 2014 [44]	BWAS	Effectiveness of MAT on workaholism	Primary prevention	
						Secondary prevention	MAT
						Tertiary prevention	
Zeijen M.E.L. et al. (2018) [53]	287 participants from the Netherlands Undefined jobs (85% highly educated); heterogeneous sectors (health care; research; educational; cultural; environmental; governmental agencies; self-employed)	Longitudinal study Two-wave design Three-month follow-up	Schaufeli et al., 2009 [34]	DUWAS	Association between workaholism, job crafting, and self-management strategies	Primary prevention	
						Secondary prevention	EI; MAT
						Tertiary prevention	

BWAS, Bergen Work Addiction Scale; CBT, cognitive behavioral therapy; DUWAS, Dutch Work Addiction Scale; EI, emotional intelligence; JDs, job demands; JRs, job resources; MAT, meditation awareness training; PSC, psychological safety climate; REBT, rational emotive cognitive therapy; WART, Work Addiction Risk Test; WA, Workaholics Anonymous; WLB, work–life balance.

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
