# Peer review of "Workaholism Prevention in Occupational Medicine: A Systematic Review"

_ijerph, 2021, doi:10.3390/ijerph18137109_

Round 1

Reviewer 1 Report

I greatly appreciate the effort to systematize the empirical evidence available to date regarding interventions useful in preventing and containing workaholic behavior. On the other hand, I think that several issues must necessarily be addressed by the authors in order to improve the paper and make it suitable for publication.

Comments to the Authors:

ABSTRACT: The "Results" section should include the time frame considered for collecting published papers.

PAGE 2, LINE 46: “Considering the lack of consistency in the scientific literature, the over-investment of workaholics may be socially acceptable and valued within companies”. This is actually due to the evidence that the most obvious characteristic of workaholics a tendency to be highly dedicated to their jobs and to devote much more time to their work than others do (Burke & Fiksenbaum, 2009), so it is a well-dressed addiction. This is the reason of the tolerance toward workaholism, rather than the lack of consistency in the scientific literature.

PAGE 2, LINE 56: “the positive consequences of workaholism identified in the literature can be explained by the confusion with the concept of work-engagement, which is commonly considered a positive form of heavy work-investment”.

Empirical results based on MTMM design comparing workers’ self-assessments and ratings of their colleagues, supported the discriminant validity between these constructs and provided evidence for the distinctive nature of these forms of working hard. See:

Mazzetti, G., Schaufeli, W. B., & Guglielmi, D. 2018. Are workaholism and work engagement in the eye of the beholder? A multirater perspective on differing forms of working hard. European Journal of Psychological Assessment, 34: 30-40

PAGE 2, LINE 68: “Preventing workaholism can be difficult considering the centrality of work in modern society”. Also, a clear identification of diagnostic criteria represents an important limitation.

PAGE 2, LINE 76: “the types of individual and collective preventive measures that could be considered to address workaholism, especially by occupational physicians”. In this statement the authors should be clearer. Are these measures implemented only by physicians or also by physicians? it is not really clear why this very specific focus is listed and why "especially." It would be helpful to have a sound rationale for this choice.

PAGE 2, LINE 85: “Seven databases were searched on March 2019 (PUBMED, PsycINFO, Web of Science, BASE, Research Gate, Science direct, Google Scholar via Publish and Perish”. The authors must specify a) why they excluded Elsevier’s Scopus, the largest database of peer-reviewed literature – scientific journals, books and proceedings; b) Why the research is two years old. I understand that it may have taken time to conduct the review of the literature found, however this should at least be stated in the limitations. You should at least report the integration with literature from the last 2 years as a direction for future research.

PAGE 2 LINE 93: I was quite surprised not to find the focus on physician-implemented measures listed among the selection criteria, as it was previously stated as a central feature of this literature review. The authors should clarify that point by deciding whether to report it throughout the manuscript or to assign it a somewhat smaller role consistently.

PAGE 3, LINE 95: “(5) studies on the prevention of workaholism (i.e., studies focusing on prevention or practical implications of results)”. The subsequent sections reveal that all three levels of prevention were included. This aspect should be specified here.

PAGE 3, LINE 105: “provide insight into the prevention of workaholism at the organizational or medical level”. Does the medical level refer to the individual level - opposed to the organizational level? Or “medical level” refers to the tertiary prevention of clinical established condition? The label “medical level” remains quite unclear.

FIGURE 1: A main limitations of the present study, able to provide a valuable contribution to the literature on interventions for workaholism prevention, lies in the process of study selection, not completely clear at the moment. In particular, the shift from 1210 studies to 155 represents a significant cut that needs to be argued more than it is.

Again, the jump from 155 records screened to 15 studies may raise some concern in the reader. The authors should describe in more detail why they were excluded.

PAGE 4, TABLE 1: the authors should introduce a column reporting the level of prevention. Furthermore, the column “definitions/diagnostic tools” should be divided into two different columns,  in order to provide a clearer picture.

PAGE 6, LINE 148: “These results revealed that work-related activities during leisure time were negatively associated with overall well-being, particularly for workaholic employees”. The authors should specify how research identified workaholic employees.

PAGE 8, LINE 277: “identifying potential workaholics based on individual and professional characteristics (e.g., self-efficacity, neuroticism and hours worked, position held)”. The label “self-efficacity” should be replaced with “self-efficacy”.

I also wonder how individual characteristics such as self-efficacy and neuroticism can be used as screening tools if no cut-off values are provided for determining if they can predict workaholism. In other words, at which threshold might the level of individual characteristics - such as self-efficacy and neuroticism - translate into workaholic behavior, thus becoming effective as a screening tool?

PAGE 7, LINE 202: “Balducci & al. refined this perspective as they demonstrated that while excessive JDs increase the risk of workaholism, workaholic behavior did not influence JDs over time”. This is quite surprising, since workaholic employees usually are prompted to create additional demands for more work (Schaufeli et al., 2008), even to the extent of allowing or even causing crisis situations (Porter, 1996)". It would be worthwhile to discuss this discrepancy between these findings: in order to feed one's compulsive behavior, it is reasonable to expect workaholics to create further pretext for working, hence unnecessary or self-imposed JDs.

See: Harpaz, I., & Snir, R. (Eds.). (2014). Heavy work investment: Its nature, sources, outcomes, and future directions. Routledge.

PAGE 7, LINE 224: “to increase social support for workaholics (i.e., increased task variety, more staff, more time to plan work, and increased teamwork)”. I am puzzled by framing task variety and time for work planning as social support measures or JRs. Please explain this choice.

PAGE 7, LINE 235: “in companies with poor PSC, JRs could be used by workaholics to foster their detrimental behavior”. Please delve deeper into the potential role of JR in fostering workaholism (it is rather confusing that both JDs and JRs report a positive association with workaholism, while they usually show opposite relationships with different kinds of employees' well-being (e.g., burnout and work engagement).

PAGE 10, LINE 358: “From a practical standpoint, they suggested that encouraging voluntary job mobility should help employees find a job that fits them best in terms of well-being and should decrease workaholism behavior”. Actually, employees with person characteristics that make them prone to workaholism may be pushed to seek organizational contexts that match their compulsion. The assumption that workaholics may be attracted to certain organizations is consistent with Attraction-Selection-Attrition theory (Schneider, 1987; Schneider et al., 1995), which claims that different types of organizations attract, select, and retain different types of people. Thus, some individuals choose to work for organizations that correspond to their own traits and values (Burke, 2001). Following this lead, Porter (1996) focused on organizational cultures that required employees to perform overwork in order to achieve success and advancement, and argued that the processes of self-selection, employee recruitment, socialization, and reward systems could forge a context in which workaholics are more likely to display their compulsive behavior than in other organizations. Hence, encouraging voluntary job mobility may not be effective in helping employees prone to become workaholics to find a job that should decrease workaholism behavior.

See:

Burke, R. J. (2001). Workaholism in organizations: the role of organizational values. Personnel Review, 30(6), 637–645.

Porter, G. (1996). Organizational impact of workaholism: Suggestions for researching the negative outcomes of excessive work. Journal of Occupational Health Psychology, 1, 70–84.

Schneider, B. (1987). The people make the place. Personnel Psychology, 40, 437–453.

Schneider, B., Goldstein, H. W., & Smith, D. B. (1995). The ASA framework: An update. Personnel Psychology, 48(4), 747–773.

PAGE 12, “GLOBAL DISCUSSION”: “However, a recent metanalysis concluded that there is no significant relation between workaholism and education level”. On the other hand, several studies explored the hp that work roles characterized by a high level of autonomy could be vulnerable to workaholism.

See:

Andreassen, C. S., Nielsen, M. B., Pallesen, S., & Gjerstad, J. (2019). The relationship between psychosocial work variables and workaholism: Findings from a nationally representative survey. International Journal of Stress Management26(1), 1.

Taris, T. W., Van Beek, I., & Schaufeli, W. B. (2012). Demographic and occupational correlates of workaholism. Psychological Reports110(2), 547-554.

PAGE 12, LINE 475: “Organizations should therefore take into consideration these potential risks induced by ICTs, especially for workaholic people”.

In this regard, several scholars identified the phenomenon of techno-addiction, as an uncontrollable “have to” pressure paired with anxiety when not using ICTs, which results in the use of them for long periods in an excessive way. See Salanova, M., Llorens, S., & Cifre, E. (2013). The dark side of technologies: Technostress among users of information and communication technologies. International Journal of Psychology48(3), 422-436.

PAGE 13, LINE: “can be linked with 498 the workaholic’s performance-based self-esteem and perfectionist personality”.

Recent findings clarify the relationship of perfectionism with positive and negative forms of work involvement (work engagement and workaholism).

See:

Mazzetti, G., Guglielmi, D., & Schaufeli, W. B. (2020). Same involvement, different reasons: How personality factors and organizations contribute to heavy work investment. International Journal of Environmental Research and Public Health, 17(22), 1-19

PAGE 13, LINE 419: “To prevent occupational exclusion and to help workaholic employees remain in employment, all preventive measures of the first and second levels presented above should also be considered at this stage of prevention”. It is especially relevant to emphasize this aspect because, overall, the literature review suggests a combination of individual and organizational antecedents in the emergence of workaholic behavior. Thus, only acting on secondary or tertiary prevention would restore the individual's state of well-being and enhance those resources acting as a protective factor against work addiction but excluding primary prevention measures can generate a vicious circle whereby the recovered individual is reintegrated into a pathological context.

PAGE 14 “LIMITATIONS”: Among the limitations, the authors should include the underrepresentation of blue-collar workers, despite the empirical results revealing no significant differences compared to other job positions.

Author Response

Please see he attachment

Reviewer 2 Report

Dear authors, 

I  would like to thank   you for giving me the opportunity to review this interesting article.

 This articles presents preventive measures against workaholism, especially in occupational medicine by a systematic literative  review.

Introduction :

The definition of workaholism  should  be clarified :Oates further defined a workaholic as a “person whose need for work has become so excessive that it creates noticeable disturbance or interference with his bodily health, personal happiness, and interpersonal relations, and with his smooth social functioning”.

The authors could add data on the prevalence of workaholism.

  The authors could add information on direct and indirect costs of workaholism to justify the interest of preventive measures.

Methods :

Some numbers are missing in Figure 1.

In the "study selection" paragraph, the authors could provide more détails on the choice of the type of studies retained in relation to the level of evidence. Stronger justification is required.

Results

  the numbering 2.1.4 should be changed to 3.1.4

Discussion : the depth of discussion is adequate

Round 2

Reviewer 1 Report

I believe that the authors have elaborated a revision of the document which has certainly improved the manuscript, now worthy of publication.